# Physical Activity Estimated by the Wearable Device in Lung Disease Patients: Exploratory Analyses of Prospective Observational Study

**DOI:** 10.3390/jcm12134424

**Published:** 2023-06-30

**Authors:** Kentaro Ito, Maki Esumi, Seiya Esumi, Yuta Suzuki, Tadashi Sakaguchi, Kentaro Fujiwara, Yoichi Nishii, Hiroki Yasui, Osamu Taguchi, Osamu Hataji

**Affiliations:** 1Respiratory Center, Matsusaka Municipal Hospital, Matsusaka 515-0073, Japan; 2Biostatistics, Yokohama City University, Yokohama 236-0004, Japan

**Keywords:** physical activity, lung disease/pulmonary disease, healthy case, wearable fitness tracker device

## Abstract

**Background.** Physical activity is a potential parameter to assess the severity or prognosis of lung disease. However, the differences in physical activity between healthy individuals and patients with lung disease remain unclear. **Methods.** The analyses in this report are a combined analysis of four cohorts, including a healthy control cohort, in a prospective study designed to evaluate wearable device-estimated physical activity in three cohorts: the lung cancer cohort, the interstitial pneumonia cohort, and the COPD cohort (UMIN000047834). In this report, physical activity in the lung disease cohort was compared with that in the healthy cohort. Subgroup analyses were performed based on age, sex, duration of wearable device use, and lung disease subtype. **Results.** A total of 238 cases were analyzed, including 216 patients with lung disease and 22 healthy cases. Distance walked and number of steps were significantly lower in the patient group compared to the healthy control group. ROC analysis for the diagnostic value of lung disease by mean distance walked and mean number of steps showed AUC of 0.764 (95%CI, 0.673 to 0.856) and 0.822 (95%CI, 0.740 to 0.905), respectively. There was a significant difference in physical activity by age, but not by gender nor by duration based on the threshold of 7 days of wearing the device. **Conclusions.** Lung disease decreases physical activity compared to healthy subjects, and aging may bias the estimation of physical activity. The distance walked or number of steps is recommended as a measure of physical activity, with a period of approximately one week and adjusted for age for future investigation.

## 1. Introduction

Physical activity is an important parameter to evaluate an individual’s general health status. Some reports have indicated the importance of physical activity in patients with lung diseases, especially COPD [1,2,3,4,5]. reports about the difference in physical activity between the patients with lung diseases and healthy cases were limited. In practice, there are even some patients with serious diseases, such as lung cancer, who show full activity that is not inferior to that of healthy individuals. Therefore, we investigated whether there is a difference in physical activity between patients and healthy individuals before discussing whether the measurement of physical activity is available to assess the severity of a specific disease. We hypothesized that there is difference in physical activity between the healthy population and lung disease population and that the physical activity value is available for evaluating the general status among lung disease population. The aim of this exploratory analysis is to investigating the hypothesis. We first identified the useful measurements with a comparison of each physical activity between the healthy population and lung disease population, and investigate the diagnostic value of physical activity for lung disease. We expected that METS and distance walked would correlate better with disease status and prognosis than step count, so we used a wearable device that could measure all of these. The data of the three cohorts and healthy control in a prospective study to evaluate disease severity by physical activity estimated by wearable device were used for the analyses as an exploratory investigation.

## 2. Materials and Methods

### 2.1. Patients

The results of this report are from exploratory analysis of the main study, which was designed to assess disease severity in three disease cohorts using a wearable device (UMIN000047834), and the aim of this analysis was to compare physical activity between healthy cases and patients with lung disease.

The main study is a prospective observational study at a single-institution, and the inclusion criteria included (1) patients diagnosed with advanced lung cancer, interstitial pneumonia or COPD, and (2) outpatients for treatment or observation at a single institution. Patients with interstitial pneumonia were enrolled regardless of the type of interstitial pneumonia, from interstitial pulmonary fibrosis to collagen-disease-related interstitial pneumonia, and patients with COPD were diagnosed in clinical practice based on pulmonary function testing. The exclusion criteria included (1) patients with unstable gait due to lower limb disease. Healthy controls without lung disease were recruited from the medical staff. The healthy group was enrolled as an exploratory group, separate from the other three cohorts in this analysis, for the purpose of exploratory validation as an exploratory study group.

### 2.2. Measurement of Physical Activity

We used the wearable device, amue link (SONY, Inc., Tokyo, Japan), which is equipped with a GPS system that can monitor real time movements without any other device, such as smart phone. To estimate the distance walked, this device adopts a unique algorithm to distinguish the method of travel, such as by vehicle or on foot. This device is also expected to calculate the actual walked distance. In this study, this device measured a total of 6 physical activity values as follows: sum and mean of (1) METS, (2) distance walked, and (3) number of steps. All patients were instructed to wear the wearable device on their wrist, following the company’s recommendation that the device on the wrist can adequately measure distance walked and steps taken. The duration of measurement was from wake-up to bedtime, with the exception of special situations such as bath time. All participants were instructed to wear the fitness tracker device for up to 14 days. The data were sent to the data center via the Internet and stored in real time with the reservation of privacy.

### 2.3. Outcome

The wearable device measurements were compared between the healthy and lung disease groups, and we estimated the AUC to discriminate the healthy from lung disease groups by physical activity in the ROC analysis. Although the aim was not to discriminate disease by physical activity, we assumed that the high accuracy of discriminating disease by physical activity, represented by a higher AUC, means that physical activity is associated with the presence or absence of respiratory disease. Comparisons of physical activity were made between patients with lung disease and healthy controls, and between the different types of lung disease. We hypothesized that the factors of age, sex, and the duration the device is worn may bias the estimation of physical activity and subgroup analyses performed based on these factors.

### 2.4. Statistics and Ethics

As described above, physical activity, as continuous values, were compared between patients with lung disease and healthy controls, and between different types of lung disease. The physical activity values were compared between the healthy group and lung disease using the Kruskal–Wallis test, and the multiple comparison tests across groups were performed using the Dunn–Bonferroni method. Statistical comparisons of the continuous values were tested with 0.05 representing a significant difference. ROC analyses were performed, and the 95% confidential intervals were calculated by SPSS version 28.0. As this is an exploratory study, sample size calculations were not performed. We obtained written informed consent from all enrolled cases and followed them prospectively in this study. The study was conducted in accordance with the principles of the Declaration of Helsinki.

## 3. Results

### 3.1. Characteristics

A total of 238 cases were analyzed, including 22 healthy cases, 119 patients with lung cancer, 51 patients with interstitial pneumonia, and 46 patients with COPD (Table 1). Written informed consent was obtained from all enrolled cases, and we observed them prospectively. For all cases, the median age was 73 years (range, 24 to 88) and the duration of device wear was 8 days. The median and the duration of wear were not significantly different in each group, but the age is significantly younger in the healthy group compared to the other groups (*p* < 0.001). The failure rate for physical activity assessment in the whole population is 5.5%. The physical activities measured by the wearable device showed a stronger correlation with each other, except for METS.

### 3.2. Healthy vs. Pulmonary Disease

Nonparametric test was performed for the measured value estimated by the wearable device between groups. As a result, the distance walked and the number of steps walked were candidate parameters as physical activities for diagnosis of lung disease. The physical activities in the patient group were statistically significantly lower compared with the healthy control group, with p value of 0.05 or less for the sum of METS, sum and mean of the distance walked, and sum and mean of the number of steps walked (Figure 1). ROC analysis for the diagnostic value of lung disease by distance walked showed an AUC of 0.764 (95%CI, 0.673 to 0.856) (Figure 2).

### 3.3. Across the Subtype of Pulmonary Disease

There was significant difference in physical activity in except with METS between healthy control group and each lung disease group (Table 2). For all lung diseases, the physical activity has trend to be decreased compared with healthy control group, and the largest difference compared with healthy group was found in lung cancer cohort, while the smallest difference was in COPD cohort (Figure 3).

### 3.4. Subgroups by Age, Gender, and the Wearing Period

Subgroup analysis by age showed that patients aged 75 years or older had less physical activity in all disease subtypes, and distance and steps walked were significantly different between younger and older patients (Figure 4). There was no difference in physical activity between men and women in all disease subtypes, except for COPD, where the sample size of women was very small. Analyses based on time worn showed that there was no difference in physical activity between 7 or fewer days worn and 8 or more days worn. This trend is consistent across lung disease subtypes, so the data suggest that one week may be a sufficient estimation period. Finally, based on these results, we hypothesized that age may be biased in discriminating the reason for lower physical activity due to disease status or aging. Therefore, we conducted an ROC analysis in the subgroup of 69 or younger cases, which resulted in an AUC of 0.732 (95%CI, 0.617 to 0.848) to 0.825 (95%CI, 0.721 to 0.929) (Figure 5).

## 4. Discussion

The analyses in this study indicated the following three important clinical questions: (1) which measurement among wearable device is appropriate to assess physical activity, (2) which factor should be adjusted, and (3) how long of a term is required to estimate physical activity? 

Our data showed a significant difference in physical activity between the patients with lung disease and healthy cases. In previous reports, Fabio Pitta et al. reported that patients with COPD were inactive compared with healthy elderly patients [6], and Benoit Wallaert et al. reported that the patients with fibrotic idiopathic interstitial pneumonia also had less activity in their daily lives compared to healthy controls [7], which is consistent with the results of our study. However, there was a significant difference in age between these two groups, which was further investigated in the subgroup analyses. 

A greater difference was found in the lung cancer group followed by interstitial pneumonia group among three cohort, from which we expect that an measurement of physical activity will be of clinical significance, especially in the two pulmonary diseases. Sofie Breuls et al. showed that the patients with interstitial pneumonia were less active compared with the patients with COPD in their propensity score analysis. This study used the daily steps to evaluate the physical activity [8]. As we can see in Figure 3, our data also showed that the physical activity, in both distance walked and steps walked, were lower in the interstitial pneumonia group compared with the COPD group. In addition, the proportion of the patients aged 75 or older was higher in the COPD group, with 47.1% in the interstitial pneumonia group and 60.9% in the COPD group, which supports the inactivity of patients with interstitial pneumonia.

There was a significant difference in the distance walked and the number of steps walked between each disease cohort and the healthy group, and these two measurements seemed to be the candidate parameters of physical activities in our study. In our study, METS is not a better candidate to evaluate the physical activity, however, the formula for estimating METS is specified for the device, therefore, the results in our study do not mean that METS is not completely excluded as a parameter of physical activity when using the other device. The previous study, as mentioned above, assessed physical activity using steps, and our data also showed that the number of steps walked was a better predictor of lower physical activity due to lung disease. Meanwhile, our data showed that mean distance walked also had potential as a candidate measure of physical activity in the lung disease population. In particular, this device, amue link, was designed to calculate the distance by using a unique algorithm to identify movement on foot, which may be the reason why the distance walked was a better predictor of physical activity in this study.

The subgroup analyses based on age, sex, or duration of device wear showed that age may be a bias for physical activity, suggesting that these measured activities require adjustment based on age but not sex. Therefore, we performed an additional subgroup analysis of cases of individuals aged 69 or younger, in which the difference in physical activity between patients and healthy controls has remained consistent, and the AUC in this subgroup is 0.732 (95%CI, 0.617 to 0.848), which is not as inferior compared to that in all cases (0.764 (95%CI, 0.673 to 0.856)). The decrease in physical activity with age has been reported in the previous reports [9,10,11], which is consistent with our results. However, the female sex has also been reported to be associated with a decreased physical activity [10], which was not confirmed in our study, probably because of the imbalance in the proportion of male and female participants. This indicates that a decreased physical activity is not only due to aging, but also to lung disease, so the assessment of physical activity in patients with lung disease should be performed in any disease cohort with adjustment for age. The subgroup based on the days of wearing the device showed that the duration of physical activity estimation may be sufficient to be approximately one week. However, the results were suggestive rather than conclusive, and further confirmation is needed for them to be associated with disease severity, even after adjusting for age. 

This study has some limitations. Failure to measure physical activity was calculated to occur at 5.5%, which may bias the analysis. There are some outliers in the data, which are assumed to be caused by errors with the device in estimating physical activity. The analyses were not prospectively designed, and the sample size of the healthy control group is small, although the sample size of the pulmonary disease group is sufficiently large, which may be a limitation. In addition, it is inevitable that the variation across disease will be a limitation of the analysis.

## 5. Conclusions

The lung disease decreased the physical activity compared with healthy cases in our study. Considering the fact that the difference in physical activity between healthy case and lung disease patients, the physical activity had potential to be reflected on the severity of the disease. We are preparing to report the association between the lung disease severity and physical activity in the next report of this study, which is the primary endpoint of the main study. Our data showed that age should be adjusted for when assessing physical activity, but the reduced physical activity in patients with lung disease was found even in the younger population. The subtype of lung disease seems to be associated with reduced physical activity. The number of steps walked was a good parameter of physical activity in our study, as shown in the previous study indicated. In addition, the distance walked was also a good indicator of daily activity when calculated by the device used in our study. We recommend using the distance or steps walked as a measure of physical activity with the period of about one week while adjusting for age for future investigations.

## Figures and Tables

**Figure 1 jcm-12-04424-f001:**
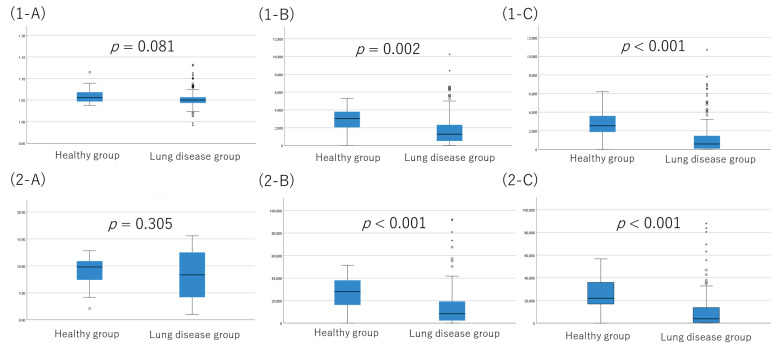
Comparison of physical activity estimated by ePRO between healthy group and pulmonary disease group. Three parameters of physical activity were shown as (**1-A**,**2-A**) METS, (**1-B**,**2-B**) distance walked, and (**1-C**,**2-C**) number of steps of (**1-A**–**1-C**) the average (on the top) and (**2-A**–**2-C**) the total (on the bottom). In a box plot, the box represents the interquartile range, with its top and bottom indicating the third and first quartiles, respectively. The whiskers extend from the box to cover the data within an area 1.5 times the size of the interquartile range. Data points outside this range are considered outliers and are represented as dots.

**Figure 2 jcm-12-04424-f002:**
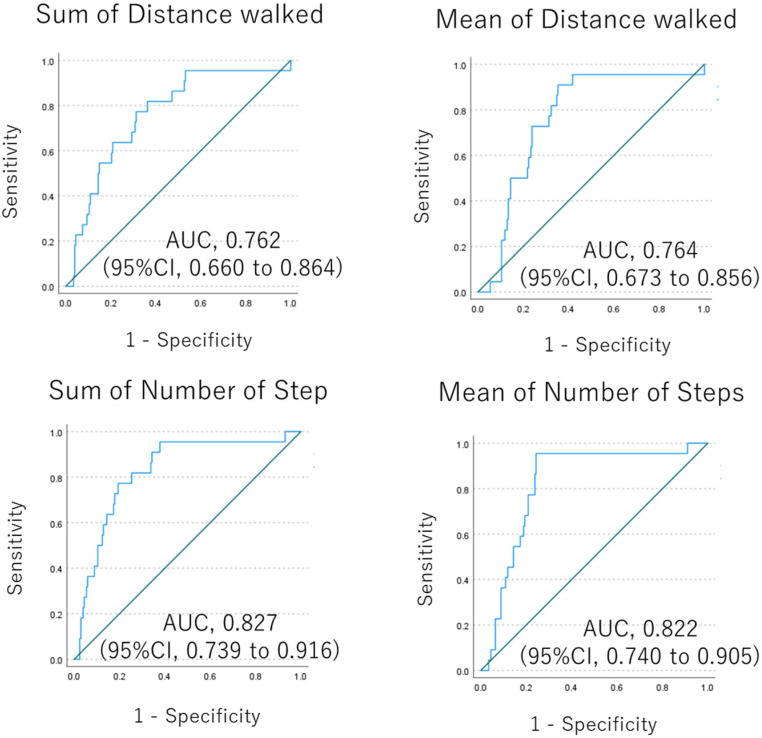
AUC for diagnosis with lung disease by physical activity in ROC analysis. Blue line is ROC curves of sum of distance walked or mean of distance walked. Green line is a reference line.

**Figure 3 jcm-12-04424-f003:**
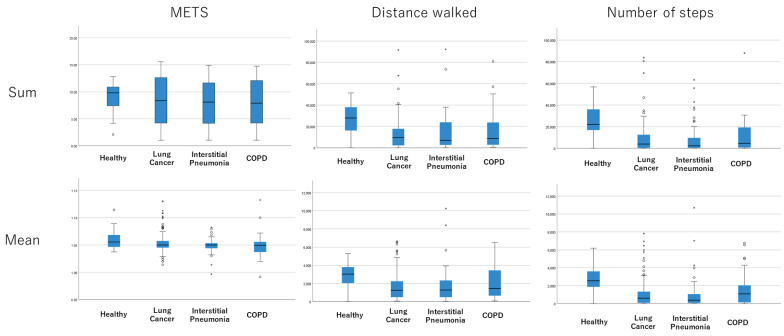
Physical activity in each lung disease subtype. In each graph, from left to right, healthy cases, lung cancer patients, interstitial pneumonia patients, and COPD patients. In a box plot, the box represents the interquartile range, with its top and bottom indicating the third and first quartiles, respectively. The whiskers extend from the box to cover the data within an area 1.5 times the size of the interquartile range. Data points outside this range are considered outliers and are represented as dots.

**Figure 4 jcm-12-04424-f004:**
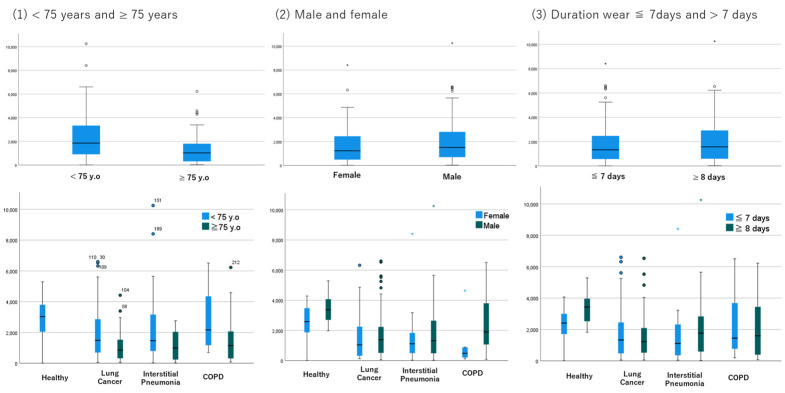
Subgroup analyses based on age, sex, and the duration worn. Comparison of mean distance walked between (**left**) <75 years and ≥75 years, (**middle**) female and male, (**right**) worn less than 7 days and more than 7 days. Each comparison based on lung disease is shown in the bottom row. In a box plot, the box represents the interquartile range, with its top and bottom indicating the third and first quartiles, respectively. The whiskers extend from the box to cover the data within an area 1.5 times the size of the interquartile range. Data points outside this range are considered outliers and are represented as dots.

**Figure 5 jcm-12-04424-f005:**
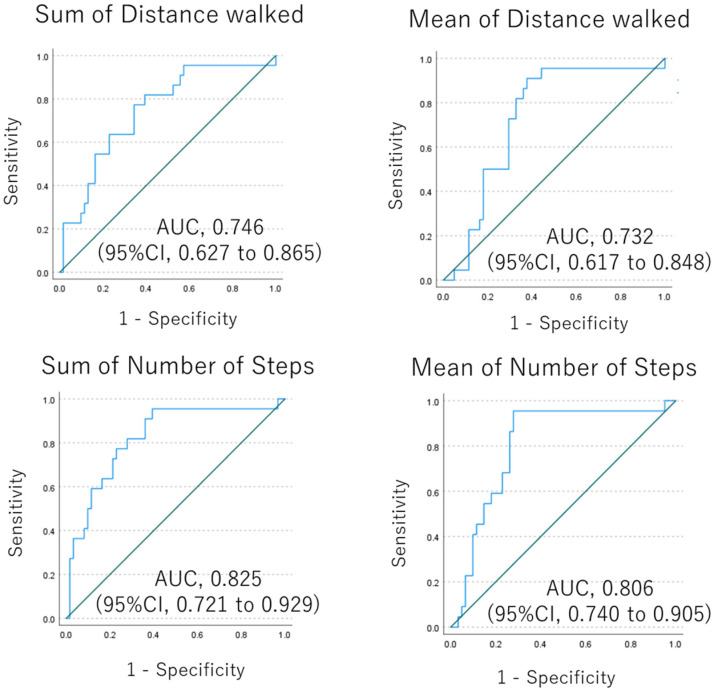
ROC analysis for diagnosis with pulmonary disease in the subgroup of cases with age less than 70. Blue line is ROC curves of sum of distance walked or mean of distance walked. Green line is a reference line.

**Table 1 jcm-12-04424-t001:** Characteristics.

	All	Healthy Control	Pulmonary Disease	Lung Cancer	Interstitial Pneumonia	COPD
N	238	22	216	119	51	46
Median age [range] ≥75 years old	73 (24–88) 99 (41.6%)	36 (24–58) 0 (0%)	74 (32–88) 99 (45.8%)	72 (32–88) 47 (39.5%)	74 (54–85) 24 (47.1%)	75 (43–83) 28 (60.9%)
Sex M/F	152/86	9/13	143/73	71/48	34/17	38/8
Wearing days, median ≤7 days	8 117 (49.2%)	9 7 (31.8%)	7 110 (50.9%)	7 60 (50.4%)	8 25 (49.0%)	7 25 (54.3%)
Failure to estimate PA	13 (5.5%)	0 (0%)	13 (%)	8 (6.7%)	0 (0%)	5 (10.9%)

Abbreviation: Physical activity, PA; Chronic obstruct pulmonary disease, COPD.

**Table 2 jcm-12-04424-t002:** Kruskal–Wallis test and multiple comparison test by Dunn-Bonferroni method for each paired groups.

Parameter of Physical Activity to Subtype of Pulmonary Disease for Null Hypothesis	Kruskal–Wallis Test	Multiple Comparison Test					
		LK vs. IP	LK vs. CO	LK vs. HE	IP vs. CO	IP vs. HE	CO vs. HE
Sum of METS	0.769	-					
Mean of METS	0.090	-					
Sum of distance walked	<0.001	0.892	0.596	<0.001	0.723	<0.001	0.002
Mean of distance walked	<0.001	0.842	0.178	<0.001	0.310	<0.001	0.006
Sum of steps walked	<0.001	0.609	0.437	<0.001	0.276	<0.001	<0.001
Mean of steps walked	<0.001	0.409	0.219	<0.001	0.083	<0.001	<0.001

Abbreviations: LK, lung cancer; CO, COPD; IP, interstitial pneumonia; HE, healthy control case.

## Data Availability

The data in this study will be available in accordance with Japanese law for personal privacy protection by request to the corresponding author.

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
