# Peer review of "Physical Activity Estimated by the Wearable Device in Lung Disease Patients: Exploratory Analyses of Prospective Observational Study"

_jcm, 2023, doi:10.3390/jcm12134424_

Round 1
Reviewer 1 Report
GENERAL COMMENTS
The authors performed an exploratory analysis of a prospective observational study to evaluate wearable device-estimated physical activity.
It is interesting. However, there are some specific comments that it is necessary to resolve.
SPECIFIC COMMENTS
INTRODUCTION
Why is essential the use of wearable devices to measure physical activity?
METHODS
It is important to include the design of study, and the inclusion/exclusion criteria of participants.
There is a lot of variability between the patients included in the three cohorts (lung cancer, interstitial pneumonia, COPD).
“Healthy controls without lung disease were recruited from the medical staff” Are these data collected of cohort studies?
What are the variables included in the study?
The statistical analysis needs to be more specific. There is another kind of analysis in the results, it didn’t describe.
Author Response
Thank you for your reviewing our manuscript. We appreciate for your comments to improve our manuscript, and we revised the manuscript according to your review as below.
Point 1. Why is essential the use of wearable devices to measure physical activity?
Response 1. Thank you for the main question of this study. We hypothesized that METS and distance walked would correlate better with disease status and prognosis than step count, so we used a wearable device that could measure all of these. We have added the above text to the Introduction.
Point 2. It is important to include the design of study, and the inclusion/exclusion criteria of participants.
Response 2. Thank you for your important suggestion. Inclusion criteria are advanced stage lung cancer, interstitial pneumonia and COPD, and patients attending the hospital as outpatients and patients with walking disability are excluded. The criteria for the main analysis are as shown above because the details are not specified. Inclusion and exclusion criteria are listed in the patient section, but we thought the text was unclear, so we corrected it for clarity. We have revised the mention of the study design to be clear. Thank you for pointing this out.
Point 3. There is a lot of variability between the patients included in the three cohorts (lung cancer, interstitial pneumonia, COPD).
Response 3. We fully agree with the reviewer's point. We believe that this study is disease specific because there are differences in the amount of physical activity in each disease. This means that it is important to evaluate the amount of physical activity for each underlying disease. In fact, although we cannot disclose it yet, we are analyzing life expectancy and have found that the prognostic value of physical activity is high for LK and IP. We have added variability as a limitation to the overall case. We appreciate you for your important comment.
Point 4. “Healthy controls without lung disease were recruited from the medical staff” Are these data collected of cohort studies?
Response 4. Thank you for your very important point, and I agree that Healthy Control was not explained well enough. The healthy controlled were enrolled just for exploratory analysis, so this cohort was not included in main analysis. We added the explanation in the Patients section of the Materials and Methods section.
Point 5. What are the variables included in the study?
Response 5. Thank you for your comment. Regarding the variables, I will answer assuming that the variable is the one that is the subject of the outcome of this exploratory study. The valuable for measurements are three (1) METS (2) Walked distance and (3) Number of step. The variables are listed in the Physical Activity Measurement section. Please let me know if there are any variables that should be added.
Point 6. The statistical analysis needs to be more specific. There is another kind of analysis in the results, it didn’t describe.
Response 6. Thank you for your important comments. We agree with you that some statistical methods are missing. We have added a sentence in the Statistics section about the statistical methods used to compare physical activity levels.
Reviewer 2 Report
Dear authors,
I have some major comments that should be incorporated:
- Improves the figure's quality (This is mandatory).
- Lines 40-43: The hypothesis should be rewritten.
- Introduction: What is the major aim of this study?
- Statistical analysis: Why do you use AUC analysis?
*** Please, give data about the sample size calculation.
*** Why use "level of physical activity" (expressed as METs)? I suggest that the intensity of PA can be described (IPAQs)
Author Response
Thank you for reviewing and providing us with meaningful suggestions.
We have responded to the reviewer's point as follows.
Point 1. - Improves the figure's quality (This is mandatory).
Thank you for your comments. I will improve the figures, especially bar graph. ROC curves can not be improved more in the aspect with resolution.
Point 2. - Lines 40-43: The hypothesis should be rewritten.
As the reviewer pointed out, we agree that the described hypothesis is not appropriate for the analysis. We revised the hypothesis according to the aim of this study as below: “We hypothesized that there is difference in physical activity between healthy population and lung disease population, and that the physical activity value is available for evaluating the general status among lung disease population.”
Point 3. - Introduction: What is the major aim of this study?
We are sorry that the aim of this study is unclear in the present manuscript. We revised the aim of study in the Introduction section as below: “The aim of this exploratory analysis is to investigating the hypothesis, and we first identified the useful measurements with comparison of each physical activity between healthy population and lung disease population, and investigate the diagnostic value of physical activity for lung disease.”
Point 4.- Statistical analysis: Why do you use AUC analysis?
Generally, the comparative analysis of continuous value between two groups is used for this analysis, but it is guessed that the value is expected to reflect continuously on the severity of the disease. In other words, the comparison between two groups is not sufficient for prove the availability of the PA value for evaluating general status. As described in test, we assumed that the high accuracy of discriminating disease by physical activity, represented by a higher AUC, means that physical activity is associated with the presence or absence of respiratory disease. AUC by ROC analysis, similar to sensitivity or specificity, is recommended for evaluating diagnostic value in the STARD guideline.
Point 5. *** Please, give data about the sample size calculation.
As this is an exploratory study, sample size calculations were not performed. Even in the main analysis, the study is conducted as a pilot study because it is difficult to hypothesize differences in physical activity. According to your comment, I have added a note that sample size calculations will not be performed.
Point 6. *** Why use "level of physical activity" (expressed as METs)? I suggest that the intensity of PA can be described (IPAQs)
We adopted METs because the Amuelink wearable device can calculate METs, which probably can be suitable value for being evaluated by wearable device.
Thank you for pointing out IPAQs. Unfortunately, in this study, we did not conduct a questionnaire survey using IPAQs, so we cannot include it in the analysis. However, we appreciate your suggestion, and we would like to consider collecting data using IPAQs for future research and conducting a comparison. Thank you for your feedback.
Round 2
Reviewer 1 Report
The manuscript need to include the ethical aspect in the methods section.
Author Response
Response to Reviwer1
Thank you for your reviewing our manuscript. We appreciate for your comments to improve our manuscript, and we revised the manuscript according to your review as below.
Point 1. Why is essential the use of wearable devices to measure physical activity?
Response 1. Thank you for the main question of this study. We hypothesized that METS and distance walked would correlate better with disease status and prognosis than step count, so we used a wearable device that could measure all of these. We have added the above text to the Introduction.
Point 2. It is important to include the design of study, and the inclusion/exclusion criteria of participants.
Response 2. Thank you for your important suggestion. Inclusion criteria are advanced stage lung cancer, interstitial pneumonia and COPD, and patients attending the hospital as outpatients and patients with walking disability are excluded. The criteria for the main analysis are as shown above because the details are not specified. Inclusion and exclusion criteria are listed in the patient section, but we thought the text was unclear, so we corrected it for clarity. We have revised the mention of the study design to be clear. Thank you for pointing this out.
Point 3. There is a lot of variability between the patients included in the three cohorts (lung cancer, interstitial pneumonia, COPD).
Response 3. We fully agree with the reviewer's point. We believe that this study is disease specific because there are differences in the amount of physical activity in each disease. This means that it is important to evaluate the amount of physical activity for each underlying disease. In fact, although we cannot disclose it yet, we are analyzing life expectancy and have found that the prognostic value of physical activity is high for LK and IP. We have added variability as a limitation to the overall case. We appreciate you for your important comment.
Point 4. “Healthy controls without lung disease were recruited from the medical staff” Are these data collected of cohort studies?
Response 4. Thank you for your very important point, and I agree that Healthy Control was not explained well enough. The healthy controlled were enrolled just for exploratory analysis, so this cohort was not included in main analysis. We added the explanation in the Patients section of the Materials and Methods section.
Point 5. What are the variables included in the study?
Response 5. Thank you for your comment. Regarding the variables, I will answer assuming that the variable is the one that is the subject of the outcome of this exploratory study. The valuable for measurements are three (1) METS (2) Walked distance and (3) Number of step. The variables are listed in the Physical Activity Measurement section. Please let me know if there are any variables that should be added.
Point 6. The statistical analysis needs to be more specific. There is another kind of analysis in the results, it didn’t describe.
Response 6. Thank you for your important comments. We agree with you that some statistical methods are missing. We have added a sentence in the Statistics section about the statistical methods used to compare physical activity levels.

Reviewer 2 Report
Ok, thanks for the authors incorporating the suggestions given.
The manuscript should be improved concerning the English language.
Ok, thanks for the authors incorporating the suggestions given.
The manuscript should be improved concerning the English language.
Author Response
Response to Reviewer2
Thank you for reviewing and providing us with meaningful suggestions.
We have responded to the reviewer's point as follows.
Point 1. - Improves the figure's quality (This is mandatory).
Thank you for your comments. I will improve the figures, especially bar graph. ROC curves can not be improved more in the aspect with resolution.
Point 2. - Lines 40-43: The hypothesis should be rewritten.
As the reviewer pointed out, we agree that the described hypothesis is not appropriate for the analysis. We revised the hypothesis according to the aim of this study as below: “We hypothesized that there is difference in physical activity between healthy population and lung disease population, and that the physical activity value is available for evaluating the general status among lung disease population.”
Point 3. - Introduction: What is the major aim of this study?
We are sorry that the aim of this study is unclear in the present manuscript. We revised the aim of study in the Introduction section as below: “The aim of this exploratory analysis is to investigating the hypothesis, and we first identified the useful measurements with comparison of each physical activity between healthy population and lung disease population, and investigate the diagnostic value of physical activity for lung disease.”
Point 4.- Statistical analysis: Why do you use AUC analysis?
Generally, the comparative analysis of continuous value between two groups is used for this analysis, but it is guessed that the value is expected to reflect continuously on the severity of the disease. In other words, the comparison between two groups is not sufficient for prove the availability of the PA value for evaluating general status. As described in test, we assumed that the high accuracy of discriminating disease by physical activity, represented by a higher AUC, means that physical activity is associated with the presence or absence of respiratory disease. AUC by ROC analysis, similar to sensitivity or specificity, is recommended for evaluating diagnostic value in the STARD guideline.
Point 5. *** Please, give data about the sample size calculation.
As this is an exploratory study, sample size calculations were not performed. Even in the main analysis, the study is conducted as a pilot study because it is difficult to hypothesize differences in physical activity. According to your comment, I have added a note that sample size calculations will not be performed.
Point 6. *** Why use "level of physical activity" (expressed as METs)? I suggest that the intensity of PA can be described (IPAQs)
We adopted METs because the Amuelink wearable device can calculate METs, which probably can be suitable value for being evaluated by wearable device.
Thank you for pointing out IPAQs. Unfortunately, in this study, we did not conduct a questionnaire survey using IPAQs, so we cannot include it in the analysis. However, we appreciate your suggestion, and we would like to consider collecting data using IPAQs for future research and conducting a comparison. Thank you for your feedback.
